# Influenza Vaccination in the Elderly in Three Cities in China: Current Status and Influencing Factors Under Different Funding Policies

**DOI:** 10.3390/vaccines13111158

**Published:** 2025-11-12

**Authors:** Rina Su, Hongting Zhao, Xiaokun Yang, Ying Qin, Jiandong Zheng, Xinyi Liu, Xinwei Du, Zhibin Peng

**Affiliations:** Division of Infectious Disease, Chinese Center for Disease Control and Prevention, Chinese Academy of Preventive Medicine, Beijing 102206, China; surina1205@outlook.com (R.S.); zhaoht@chinacdc.cn (H.Z.); yangxk@chinacdc.cn (X.Y.); 17839974720@163.com (X.L.); duxinwei1125@163.com (X.D.)

**Keywords:** elderly, influenza vaccine, analysis of influencing factors, vaccination services

## Abstract

**Background**: Influenza is a major health threat to the elderly in China. Despite this, influenza vaccination rates still remain low and vary across regions that have different funding policies. In this study, we compare the vaccination status and influencing factors among older adults under the free, partial reimbursement, and self-paid vaccination strategies. **Methods**: Three cities with free, partial reimbursement, and self-paid influenza vaccination policies were selected. A cross-sectional, anonymous survey was then conducted. A total of 2265 elderly individuals aged 60 years and above were recruited using probability proportionate to size sampling. A standardized questionnaire was used during face-to-face interviews to collect data regarding the influenza vaccination status and influencing factors. The statistical analyses included chi-square tests, a multivariate logistic regression, and random forest models. **Results**: Among the 2265 participants (free policy region: *n* = 426; partial reimbursement region: *n* = 633; self-paid region: *n* = 1206), vaccination rates during the 2023–2024 season were significantly higher in the free policy region (53.29%) than in the partial reimbursement (20.85%) and self-paid (13.60%) regions (*p* < 0.001). The intention to vaccinate for the 2024–2025 season was also highest in the free policy region (68.78%), followed by partial reimbursement (47.71%) and self-paid (37.15%) regions (*p* < 0.001). This result demonstrated the same trend as the vaccination behavior. Cues to action (e.g., healthcare worker or family member recommendations) positively influenced vaccinations across all of the regions. In the self-paid region, perceived barriers, such as vaccine cost and side effect concerns, significantly reduced both behaviors and the next-season intention to vaccinate. Healthcare worker recommendations were key positive factors, while misconceptions and costs were major barriers to vaccination. **Conclusions**: Vaccination rates varied significantly across regions with different influenza vaccine subsidy policies. The free policy region demonstrated the highest coverage rate, while the self-paid region exhibited the lowest, suggesting that financial policies are a key determinant of vaccination uptake. Furthermore, free vaccination policies were associated with improved influenza vaccine knowledge among the elderly. Analysis of other influencing factors revealed that healthcare workers’ recommendations played a crucial role across all policy regions, though their impact on current-season vaccination behavior and next-season vaccination intention differed by subsidy context. Further studies are needed to explore the best approaches for optimizing region-specific subsidy strategies for promoting influenza vaccination among the elderly in China.

## 1. Introduction

Influenza is an acute respiratory infection caused by influenza viruses, and it represents a continuous threat to global public health. Seasonal influenza affects 5–15% of the population annually and leads to 3–5 million severe cases and 290,000–650,000 respiratory-related deaths worldwide [1]. The elderly face an increased risk of severe outcomes, particularly those with declining immunity and a high prevalence of chronic diseases. Research has indicated that unvaccinated elderly people aged ≥65 years experience higher influenza and mortality rates than younger populations [2,3,4,5,6].

The World Health Organization has recommended the prioritization of influenza vaccination for the elderly, but the influenza vaccination rate among older Chinese adults remains significantly lower than that in developed countries, where rates typically range from 30% to 65% [7]. Data have indicated that vaccination rates among adults aged 60 and older were only 1.57%, 3.03%, 3.75%, and 4.16% between 2019 and 2023 [8]. Moreover, the implementation of a nationwide free influenza vaccination policy represents a significant challenge due to China’s large elderly population and uneven regional economic development. As a result, regionally tailored vaccination strategies have been adopted; these strategies align with the national “Healthy China 2030” initiative’s goal of improving preventive care for the aging population [9]. According to a 2019 survey by the Chinese Center for Disease Control and Prevention (CDC) [10], among 88 districts/counties across 13 provinces that provided free influenza vaccinations, the coverage rate reached 29.3%. In 85 districts/counties spanning seven provinces where partial reimbursement was available through health insurance, the vaccination rate was 0.85%. In areas where vaccination remained voluntary and self-paid, the rate was only 0.65%. These findings highlight not only disparities in vaccination policies and coverage across regions but also the need for context-specific research. Previous studies conducted in specific Chinese provinces or cities have identified several key factors that influence influenza vaccination among the elderly. These factors include generally low levels of knowledge regarding influenza and its vaccine [11], prevalent perceived barriers such as cost and concerns regarding safety [12,13], and the crucial role of healthcare worker recommendations [14]. However, these studies were often limited to single provinces or specific policy contexts. A critical gap remains in the direct comparison of how these factors, particularly within a theoretical framework like the Health Belief Model (HBM), differentially influence vaccination behaviors and the next-season vaccination intention across districts concurrent with operational funding policies at the national level. The aim of this study is to fill this gap by systematically investigating and comparing the vaccination status and its cognitive determinants among the elderly under free, partial reimbursement, and self-paid policies, establishing target intervention strategies.

## 2. Materials and Methods

This multi-center, cross-sectional study was conducted between May and August 2024 to investigate the influenza vaccination status, the next-season vaccination intention, and associated factors (based on the HBM) among the elderly under different funding policies. The study received approval from the Institutional Review Board of the Chinese CDC (Protocol Code 202320).

### 2.1. Study Population

Jiaxing (Zhejiang Province), Chengdu (Sichuan Province), and Yichang (Hubei Province) were selected as survey sites based on their respective vaccine cost policies for the elderly aged ≥60 years [15]: the free policy region (i.e., the influenza vaccine cost is reimbursed by the local government financial department, with no out-of-pocket expenses for eligible elderly), the partial reimbursement policy region (i.e., a fixed proportion of the vaccine cost is subsidized by the Basic Social Medical Insurance program, and the remaining amount is borne by the elderly themselves), and the self-paid policy region (i.e., the vaccine cost is paid by participants with no financial or insurance support).

A two-stage sampling method that employed the probability proportionate to size sampling method was implemented. Initially, communities within the three cities were categorized into three strata: urban, urban–rural fringe (i.e., transitional areas between urban and rural settings), and rural areas. Communities were sampled as units, and all communities were organized and numbered according to their administrative division codes. The cumulative population of elderly individuals (aged ≥60 years) across all of the communities was calculated. A sampling interval was derived by dividing this cumulative total by the planned number of communities to be selected. A random number was chosen within the first interval, and communities were then systematically selected based on this starting point and the sampling interval. Communities were then selected using random numbers. Within the selected communities, simple random sampling was performed using the household registration list to identify potential subjects. Individuals who did not meet the inclusion criteria or who declined participation were sequentially replaced by the next eligible individual on the sampling list. The inclusion criterion for the survey subjects was as follows:
(1)residents aged 60 and above who had lived locally for ≥6 months.

The exclusion criteria included the following:(1)individuals who did not agree to participate in the survey;(2)individuals with contraindications to the influenza vaccine; and(3)individuals who were unable to comprehend a simple, standardized screening question used to assess cognitive eligibility (e.g., “Do you plan to get a flu shot next year?”). Participants providing an irrelevant or incomprehensible response were excluded.

### 2.2. Sample Size

Using the multivariate logistic regression sample size calculation method, the sample sizes required for the free policy, partial reimbursement policy, and self-paid policy regions were calculated to be 420, 480, and 1155 subjects, respectively [16]. The sample size calculation equation is as follows:
(1)$$n=\frac{{\left\{Z_{1-a/2}[p(1-p)/B{]}^{1/2}+Z_{1-\beta }[p_{0}(1-p_{0})+p_{1}(1-p_{1})(1-B)/B{]}^{1/2}\right\}}^{2}}{\left(p_{0}-p_{1}{)}^{2}(1-B\right)}$$
(2)$$n_{p}=\frac{n_{1}}{1-R_{1,234\ldots p}^{2}}$$

The sample sizes were calculated for both the univariate (Equation (1)) and multivariate (Equation (2)) logistic regression analyses. Key parameters included a significance level (α) of 0.05 and a statistical power (1 − β) of 0.90.

The parameter estimates were derived from prior research. The vaccination rates (p_0_, p_1_) and exposure proportions (B) were based on a Chinese CDC survey and a Hong Kong study [17] that identified “belief in influenza vaccine effectiveness” (OR = 12.18) as a key predictor (used as variable X_1_). The coefficient of determination (R^2^) for the multivariate equation was set to 0.4, a conservative estimate that accounted for potential correlations among the independent variables. This consequently increased the calculated sample size.

The calculations were performed using the SSizeLogisticBin function in the powerMediation package (R version 4.1.0), with an additional 20% allowance for sample attrition.

(1)Full-subsidy region: p_0_ = 10%, p_1_ = 57.5%, B = 0.3 → Initial *n* ≈ 140(2)Partial reimbursement region: p_0_ = 5%, p_1_ = 39%, B = 0.2 → Initial *n* ≈ 160(3)Full self-payment region: p_0_ = 3%, p_1_ = 27.4%, B = 0.1 → Initial *n* ≈ 385

To account for the potential urban–rural disparities, stratification was planned within each region (urban, suburban, and rural). Thus, the total sample size required was (140 + 160 + 385) × 3 = 2055.

### 2.3. Survey Methods and Content

An anonymous standardized questionnaire was used [17], in this study, in addition to face-to-face interviews conducted by uniformly trained investigators. Verbal informed consent was obtained from participants prior to the survey. The questionnaire was composed of three sections (a total of 51 items). The first section addressed general demographic characteristics, which included age, gender, residence, education level, type of partial reimbursement, economic income, and basic health status, including the presence of chronic diseases (with simple hypertension excluded). The second section focused on the respondent vaccination status for pneumococcus, COVID-19, and influenza during the 2023–2024 season, as well as the next-season vaccination intention (August 2024–April 2025). The vaccination status and next-season vaccination intention were coded as binary variables for the statistical analysis: responses of “Yes” were coded as 1, and responses of “No” or “Not sure” were coded as 0. The third section utilized a Health Belief Model scale to assess factors influencing influenza vaccination among the elderly. This included six dimensions (a total of 36 items) that employed five-point Likert scales (from 1 = “strongly agree” to 5 = “strongly disagree”: ① knowledge of basic facts regarding influenza and influenza vaccines (3 questions); ② perceived influenza susceptibility (5 questions); ③ perceived influenza severity (i.e., beliefs about the seriousness of influenza and its potential complications) (5 questions); ④ perceived benefits of vaccination (5 questions); ⑤ perceived barriers to vaccination (11 questions); ⑥ cues to action (i.e., external triggers that prompt vaccination, such as advice from healthcare workers, reminders from family, or media campaigns) (7 questions). The verbal informed consent script and the standardized questionnaire are provided in the Appendix A.

### 2.4. Statistical Analysis

Data from questionnaires were double-entered using Epi Data 13.1 and organized using Excel. All statistical analyses were conducted using R version 4.4.1. The descriptive statistics and chi-square tests were used to summarize and compare the demographic characteristics and vaccination rates across the three regions. The HBM scale scores were compared using t-tests and one-way analyses of variance. Multivariate logistic regression models were employed to identify factors associated with the influenza vaccination. Prior to regression, the multicollinearity among independent variables was assessed using the variance inflation factor (VIF). All VIF values were below 1.61, indicating no substantial multicollinearity. The model’s overall fit was significant (likelihood ratio test, *p* < 0.001) with a McFadden’s pseudo R^2^ of 0.164. This suggested a moderate explanatory power. The model demonstrated good overall accuracy (81.7%) and specificity (93.9%), but it had a lower sensitivity (44.1%). The area under the receiver operating characteristic (ROC) curve (AUC) was 0.785. This result indicated an acceptable discriminative ability. Finally, a random forest model was implemented to evaluate and rank the importance of each questionnaire item. Variable importance was ranked using the mean decrease in the Gini index, with higher values indicating greater importance for classifying the vaccination status/intention to vaccinate. The model’s performance was validated using the out-of-bag (OOB) error estimation. The stabilities of the variable importance rankings were confirmed by running the model over 100 iterations using different random seeds. This approach demonstrated consistent results for the top factors (e.g., mean accuracy: 80.6% ± 0.9%; mean AUC: 0.778 ± 0.016). A 10-fold cross-validation was also performed, and it yielded an average accuracy of 77.4% (±11.4%) and an average AUC of 0.67 (±0.032). These results indicated that the model reliably identified the most important variables, despite some expected minor fluctuations in performance on unseen data.

## 3. Results

### 3.1. Demographic Characteristics

Nine of the 2274 surveys were excluded due to a lack of basic information or duplicate records. This resulted in 2265 valid responses for analysis. Of these valid responses, 426 were collected from the free policy region, 633 from the partial reimbursement policy region, and 1206 from the self-paid policy region. Significant differences were observed across the various demographic factors that included gender, age, living arrangements, education, medical insurance status, income, and personal and family illness histories (Table 1). Notably, the chronic disease prevalence (excluding simple hypertension) was highest in the free policy region (45.31%), followed by the partial reimbursement (40.28%) and self-paid regions (38.97%), although this difference did not reach statistical significance (*p* = 0.072). In contrast, a significantly higher proportion of respondents in the self-paid policy region were female (58.21%) and aged 60–69 years (39.97%). The partial reimbursement region had a greater number of respondents with a college education, whereas the free policy region reported higher income levels. The specific data are presented in Table 1.

### 3.2. Vaccination Status and Influenza Vaccination Intention

Significant differences were observed during the 2023–2024 season in the influenza vaccination status among the three regions (*p* < 0.05). The self-reported vaccination rate was highest in the free policy region (53.29%), followed by the partial reimbursement region (20.85%), and lowest in the self-payment group (13.60%). The absolute difference in the vaccination rates between the free and self-paid regions was 39.69%, underscoring the profound impact of funding policy. The free policy region exhibited the highest next-season intention to vaccinate (68.78%), followed by the partial reimbursement region (47.71%) and the self-paid policy region (37.15%), with significant differences (*p* < 0.05) for next-season vaccination intention (August 2024–April 2025).

Significant differences were also found in the pneumococcal vaccination status among the three regions (*p* < 0.05). The partial reimbursement region had the highest self-reported vaccination rate (49.45%), followed by the self-paid policy region (7.46%) and the free policy region (2.11%). The self-reported COVID-19 vaccination rates were similar across groups (*p* > 0.05) and they all exceeded 90%. The specific data are presented in Table 2.

### 3.3. Health Belief Model Scores

The HBM scores for the influenza vaccination among the elderly in the three regions demonstrated significant differences in all dimensions (all *p* < 0.05) (Table 3). Subjects of the free policy region had the highest scores in basic influenza knowledge (11.28 ± 1.86), perceived susceptibility (16.70 ± 3.84), perceived vaccination benefits (20.23 ± 3.41), and cues to action (23.07 ± 4.85). In addition, they reported the lowest perceived barriers (21.72 ± 6.20). Conversely, subjects in the self-paid policy group displayed not only the strongest perceived severity (18.18 ± 3.65) but also the highest perceived barriers (25.26 ± 5.25). The partial reimbursement region scores were intermediate. Further analysis of perceived barriers mean scores across the three regions revealed that the greatest barrier in both the self-paid policy and partial reimbursement regions was vaccination cost (self-paid: 3.024 ± 1.206; partial reimbursement: 3.254 ± 1.176). This was followed by concerns regarding vaccine side effects (self-paid: 2.921 ± 1.028; partial reimbursement: 2.573 ± 0.964). In contrast, the primary barriers in the free policy region were transportation inconvenience (2.31 ± 1.294) and injection fears (2.012 ± 1.048). An analysis of variance further indicated that the vaccination cost barrier exhibited the most significant regional differences (F = 140.7, *p* < 0.001).

### 3.4. Multivariate Logistic Regression Analysis

The multivariate logistic regression analysis identified two consistent factors in all three policy regions. The cues to action significantly promoted both the current-season vaccination and next-season vaccination intention (all ORs > 1, *p* < 0.05), and the perceived susceptibility positively influenced the next-season vaccination intention (ORs ranging from 1.180 to 1.258, *p* < 0.001). Basic influenza knowledge showed no significant effect in any region (all *p* > 0.05).

Beyond these commonalities, each group exhibited distinct patterns. In the free policy region, the perceived severity demonstrated a negative correlation with current-season vaccination (OR = 0.914, *p* = 0.020). In the partial reimbursement region, the perceived benefits positively influenced the next-season vaccination intention (OR = 1.111, *p* = 0.024). In the self-paid policy region, perceived susceptibility positively affected both current-season vaccination (OR = 1.066, *p* = 0.029) and next-season willingness (OR = 1.078, *p* = 0.001). Most notably, this region uniquely demonstrated that perceived barriers significantly reduced both current-season vaccination (OR = 0.944, *p* = 0.001) and next-season willingness (OR = 0.958, *p* = 0.002). Detailed data are presented in Table 4.

### 3.5. Random Forest Model Analysis

Random forest models were employed to evaluate the importance of the HBM constructs in predicting the vaccination behavior and next-season vaccination intention. The model parameters were optimized to minimize the OOB error rate. For the free policy region, the optimal OOB errors were 29.64% (mtry = 15, ntree = 974) for current-season vaccination and 16.67% (mtry = 7, ntree = 4289) for the next-season vaccination intention. The corresponding values were 17.07% (mtry = 4, ntree = 2632) and 25.75% (mtry = 5, ntree = 1211) for the partial reimbursement region, and 12.40% (mtry = 7, ntree = 737) and 23.88% (mtry = 4, ntree = 1447) for the self-paid policy region.

Data from the questionnaire items were included in the random forest model, and their importance was ranked using the average Gini decrease in descending order to analyze the vaccination behavior and next-season vaccination among the elderly in the different financing policy regions. The top 10 most influential factors in the three regions are illustrated in Figure 1, Figure 2 and Figure 3. The left portion is the vaccination status of the current season and the right portion is the next-season vaccination intention.

Healthcare worker and family member recommendations ranked among the top influential factors for both the current-season vaccination behavior and the next-season vaccination intention in the free policy region. In contrast, recommendations (including those from healthcare workers and family members) exerted no significant impact on the current-season vaccination in the partial reimbursement and self-paid policy regions. They only influenced the next-season vaccination intention. Additionally, the cost factor was included in the top 10 influential factors for either the current-season vaccination behavior or the next-season vaccination intention, but did not rank first in the partial reimbursement and self-paid policy regions.

## 4. Discussion

The results of this study clearly demonstrated that China’s influenza vaccination rates and next-season vaccination intention among the elderly were significantly influenced by financing policies, with notable disparities observed across regions with free, partial reimbursement, and self-paid policies.

During the 2023–2024 season, the vaccination rate in the free policy region reached 53.29%, while the next-season vaccination intention in the 2024–2025 season was 68.78%. Both were substantially higher than those of the partial reimbursement region (20.85% and 47.71%, respectively) and the self-paid policy region (13.60% and 37.15%, respectively). This gradient agreed with a 2019 survey conducted by the Chinese CDC [9] that reported higher coverage in the free policy areas (29.3%) than in the partial reimbursement (0.85%) and self-paid areas (0.65%). Additionally, studies from Beijing and Zhejiang have shown increased vaccination rates following the implementation of free elderly influenza vaccination policies [15,19]. Unlike the trend for influenza vaccines, pneumococcal vaccination was highest in the partial reimbursement region. One of the possible reasons may be that the reimbursement policy in this area usually covers all the non-National Immunization Program (NIP) vaccines, which include the influenza vaccine, pneumococcal vaccine, etc., so the vaccination cost in the reimbursement policy region might be lower for non-National Immunization Program (NIP) vaccines, which might reduce the cost barriers. In contrast, the free policy region’s subsidy was only specific to the influenza vaccine. The different patterns between the two vaccines also support the conclusion that financing policies are a key determinant of vaccination uptake. The findings clearly indicate that such policies significantly influence elderly vaccination behavior and intention, with both free and partial reimbursement models contributing to improved rates for the vaccines they subsidize.

In addition, the promotion of free influenza vaccination policies had an indirect positive impact on the elderly’s vaccine knowledge and health beliefs. An analysis that used the HBM score indicated that elderly people who resided in the free policy region scored highest in basic influenza knowledge, perceived susceptibility, perceived vaccination benefits and cues to action, while they exhibited the lowest perceived barriers. In contrast, individuals who resided in the self-paid policy and partial reimbursement regions had prominent perceived barriers, primarily vaccine cost. These results agreed with observations from Beijing, where the implementation of a free vaccination policy for the elderly was followed by an increase in both influenza and vaccine knowledge scores [20]. Vaccine subsidy policies directly reduce out-of-pocket costs. Furthermore, through policy promotion and social mobilization, they may indirectly enhance vaccine knowledge among the elderly, thereby fostering a more favorable health belief structure. Furthermore, the multivariate logistic regression and random forest model results showed that cues to action represented a critical dimension that influenced influenza vaccination among the elderly, and healthcare worker and family member recommendations emerged as the most impactful factors. However, this influence varied significantly across regions with different policies. In the free policy region, healthcare worker recommendations were among the top two determinants for both the current-season vaccination behavior and next-season intention. This result was consistent with numerous studies that have indicated that healthcare worker recommendations are among the most effective strategies for increasing vaccination rates among the elderly [21,22]. Furthermore, prior research has suggested a positive dose–response relationship, where increased advice by healthcare workers is associated with higher vaccination rates [23]. This underscores that under policy coverage, healthcare worker recommendations can be directly translated into vaccination action.

Conversely, in the partial reimbursement and self-paid regions, healthcare worker recommendations significantly influenced the next-season vaccination intention but showed no significant association with the current-season vaccination behavior, revealing a “current-season intervention gap.” Moreover, the cost factor was identified as the most significant vaccination barrier; however, when all data from the questionnaire items across the different HBM dimensions were considered together in the random forest model, cost did not rank among the top five most important factors. This result was consistent with a study performed by Kroneman et al. on unvaccinated influenza populations in Europe. They found that vaccine cost was not the primary determinant in Sweden; rather, the key factor was whether individuals received healthcare worker recommendations [24]. Additionally, relevant studies conducted in Lhasa (Tibet Autonomous Region) [25], Foshan (Guangdong Province) [26] and Hangzhou (Zhejiang Province) [27] in China have demonstrated that healthcare worker recommendations were major influencing factors, with an impact comparable to or greater than that of vaccine cost.

This study has several limitations. First, the inclusion of only one city per funding policy may have limited the generalizability of our findings and might not fully represent the diversity within similar policy contexts across China (e.g., ethnic minority areas and economically advanced eastern coastal regions). Second, although the total sample size was substantial, its distribution across the three distinct policy regions and the subsequent sub-analyses may have limited the statistical power of some comparisons. Third, the data collected were based on a cross-sectional survey that reflected only the vaccination status and influencing factors for the 2023–2024 season. Consequently, it is difficult to elucidate the causal relationships between vaccination behavior and its influencing factors.

The findings of this study highlighted several promising avenues for future research. First, longitudinal studies are warranted to track the dynamic evolution of vaccination behavior and its determinants over multiple influenza seasons, particularly in response to changes in subsidy policies. This would help establish causal relationships and assess the long-term impact of policy interventions. Second, future investigations should expand the geographical scope to include a more diverse range of regions, such as ethnic minority areas and economically developed eastern coastal cities, to enhance the generalizability of the findings and explore the influence of socio-cultural and regional economic factors. Third, qualitative or mixed-methods research would provide deeper insights into the underlying reasons for vaccine hesitancy and the decision-making processes of the elderly, especially in self-paid and partial reimbursement contexts. Furthermore, interventional studies that evaluate the effectiveness of specific strategies, such as structured recommendation protocols for healthcare workers, targeted health communication campaigns, digital health interventions (e.g., chatbot-based education) [28], or integrated vaccination service delivery models are required to translate these results into actionable public health practices.

## 5. Conclusions

This study was conducted across several Chinese cities, and it has underscored the profound impact of financial subsidy policies on influenza vaccination among the elderly. Our findings indicated that free vaccination policies were associated with coverage rates approximately four times higher than those in self-paid regions. Beyond financial mechanisms, the findings highlighted the critical role of socio-cognitive factors, with healthcare worker recommendations emerging as a pivotal cue to action across all policy contexts.

These insights suggest that a multi-pronged strategy that combines the strategic expansion of financial support, the strengthening of proactive recommendation systems within healthcare services, and targeted communication to address specific perceptual barriers holds promise for substantially improving vaccine uptake. Future longitudinal and more geographically diverse studies are required to confirm these relationships and explore the causal mechanisms that underlie them. Nonetheless, this study has provided valuable evidence to inform targeted public health strategies aimed at protecting the health of China’s aging population.

## Figures and Tables

**Figure 1 vaccines-13-01158-f001:**
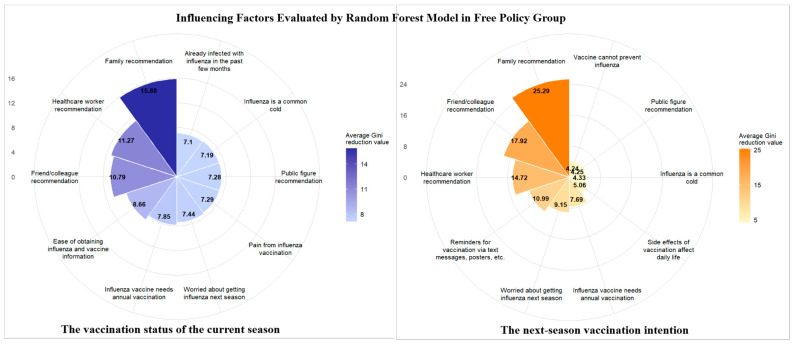
Influencing factors were evaluated using the random forest model in the free policy region.

**Figure 2 vaccines-13-01158-f002:**
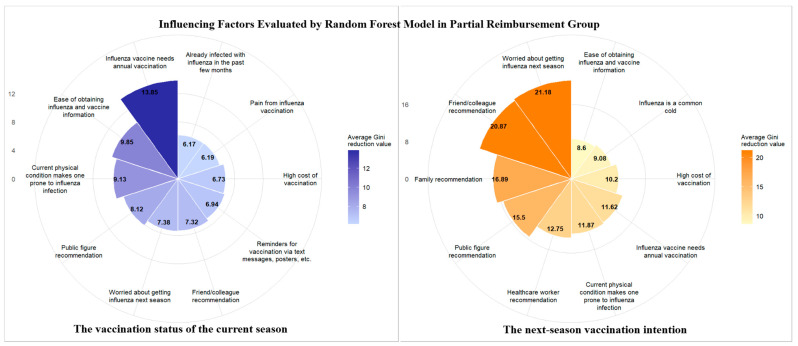
Influencing factors evaluated using the random forest model in the partial reimbursement region.

**Figure 3 vaccines-13-01158-f003:**
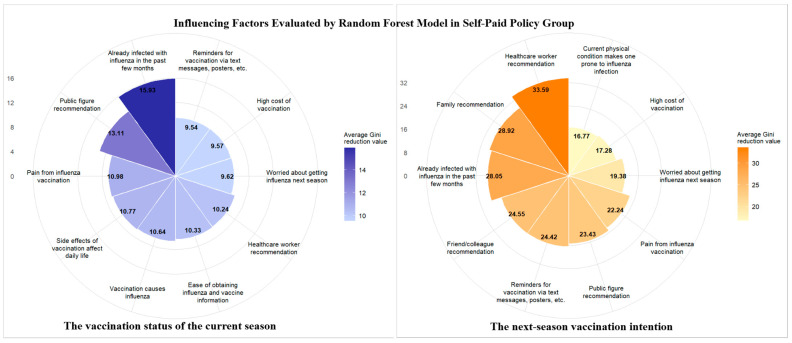
Influencing factors were evaluated using the random forest model in the self-paid policy region.

**Table 1 vaccines-13-01158-t001:** General Demographic Characteristics of Subjects of three groups.

Variable	Free Policy Region (*n* = 426)	Partial Reimbursement Region (*n* = 633)	Self-Paid Policy Region (*n* = 1206)	χ^2^ Value	*p* Value
**Region**				0.026	0.999
Urban	145 (34.04%)	214 (33.81%)	406 (33.67%)		
Urban–rural fringe	141 (33.10%)	209 (33.02%)	400 (33.17%)		
Rural	140 (32.86%)	210 (33.18%)	400 (33.17%)		
**Gender**				6.966	0.031
Male	208 (48.83%)	288 (45.50%)	504 (41.79%)		
Female	218 (51.27%)	345 (54.50%)	702 (58.21%)		
**Age (years)**				74.530	<0.001
60–69	145 (34.04%)	224 (35.36%)	482 (39.97%)		
70–79	192 (45.07%)	258 (40.76%)	449 (37.23%)		
≥80	89 (20.89%)	151 (23.88%)	275 (22.80%)		
**Living situation**				29.604	<0.001
Living alone	37 (8.69%)	59 (9.32%)	143 (11.86%)		
Living only with spouse	208 (48.83%)	283 (44.71%)	632 (52.40%)		
Living only with children	76 (17.84%)	117 (18.48%)	151 (12.52%)		
Living with spouse and children	103 (24.18%)	173 (27.33%)	280 (23.22%)		
Living in nursing home	2 (0.47%)	1 (0.16%)	0 (0.00%)		
**Education level**				275.510	<0.001
No schooling	149 (34.98%)	140 (22.12%)	90 (7.46%)		
Primary school	166 (38.97%)	235 (37.12%)	393 (32.59%)		
Junior high school	76 (17.84%)	138 (21.80%)	441 (36.57%)		
Technical secondary/high school	24 (5.63%)	74 (11.69%)	233 (19.32%)		
College	7 (1.64%)	30 (4.74%)	31 (2.57%)		
University (and above)	4 (0.94%)	16 (2.53%)	18 (1.49%)		
**Medical insurance status**				65.785	<0.001
Basic Social Medical Insurance for Urban Employees	246 (57.75%)	381 (60.19%)	520 (43.12%)		
Basic Social Medical Insurance for Urban and Rural Residents	178 (41.78%)	243 (38.39%)	664 (55.06%)		
Government Medical Scheme	0 (0.00%)	0 (0.00%)	5 (0.41%)		
No medical insurance	2 (0.47%)	8 (1.26%)	12 (1.00%)		
Other	0 (0.00%)	1 (0.16%)	5 (0.41%)		
**Average monthly family income ***				151.140	<0.001
<$138/month	0 (0.00%)	0 (0.00%)	0 (0.00%)		
$138–$414/month	27 (6.34%)	77 (12.16%)	234 (19.40%)		
$414–$690/month	114 (26.76%)	275 (43.44%)	467 (37.56%)		
$690–$1379/month	114 (26.76%)	128 (20.22%)	237 (19.65%)		
≥$1.379/month	88 (20.66%)	64 (10.11%)	50 (4.15%)		
**Presence of chronic diseases (excluding simple hypertension)**				5.270	0.072
Yes	193 (45.31%)	255 (40.28%)	470 (38.97%)		
No	233 (54.69%)	378 (59.72%)	736 (61.03%)		
**Co-resident family members with illnesses(excluding simple hypertension)**				7.394	0.025
Yes	125 (29.34%)	148 (23.38%)	279 (23.13%)		
No	301 (70.66%)	485 (76.62%)	927 (76.87%)		

* The incomes are expressed in US dollars using the 2025 exchange rate (1 US$ = 7.1161 CNY) [18].

**Table 2 vaccines-13-01158-t002:** Vaccination status and the next-season intention to vaccinate among subjects in the three groups.

Variable	Free Policy Region (*n* = 426)	Partial Reimbursement Region (*n* = 633)	Self-Paid Policy Region (*n* = 1206)	χ^2^ Value	*p* Value
**Influenza vaccination status**				283.673	<0.001
Yes	227 (53.29%)	132 (20.85%)	164 (13.60%)		
No	188 (44.13%)	483 (76.30%)	989 (82.01%)		
Not sure	11 (2.58%)	18 (2.84%)	53 (4.39%)		
**Next-season influenza vaccination intention**				127.759	<0.001
Yes	293 (68.78%)	302 (47.71%)	448 (37.15%)		
No	133 (31.22%)	331 (52.29%)	758 (62.85%)		
**Pneumococcal vaccination status**				593.806	<0.001
Yes	9 (2.11%)	313 (49.45%)	90 (7.46%)		
No	380 (89.20%)	295 (46.60%)	1062 (88.06%)		
Not sure	37 (8.69%)	25 (3.95%)	54 (4.48%)		
**COVID-19 vaccination status**				1.942	0.767
Yes	392 (92.02%)	587 (92.73%)	1129 (93.62%)		
No	22 (5.16%)	32 (5.06%)	55 (4.56%)		
Not sure	12 (2.82%)	14 (2.21%)	22 (1.82%)		

**Table 3 vaccines-13-01158-t003:** HBM scores for the influenza vaccination among the elderly in the three financing regions.

Dimension	Free Policy Region (*n* = 426)		Partial Reimbursement Region (*n* = 633)		Self-Paid Policy Region (*n* = 206)		F Value	*p* Value
	Mean	SD	Mean	SD	Mean	SD		
Basic influenza knowledge	11.28	1.86	9.99	1.74	10.52	1.99	57.862	<0.001
Perceived susceptibility	16.70	3.84	15.37	3.25	15.85	3.61	17.923	<0.001
Perceived severity	17.35	4.06	17.06	3.65	18.18	3.65	21.223	<0.001
Perceived vaccination benefit	20.23	3.41	18.69	3.04	19.30	3.55	26.155	<0.001
Perceived vaccination barriers	21.72	6.20	24.38	4.79	25.26	5.25	69.604	<0.001
Action cues	23.07	4.85	20.48	3.88	19.52	4.49	102.520	<0.001

**Table 4 vaccines-13-01158-t004:** Multivariate logistic regression analysis of the influenza vaccination among the elderly in the three policy regions.

		Free Policy Region (*n* = 426)				Partial Reimbursement Region (*n* = 633)				Self-Paid Policy Region (*n* = 1206)			
Dependent Variable	Independent Variable	β	Wald χ^2^ Value	Adjusted OR (95% CI)	*p* Value	β	Wald χ^2^ Value	Adjusted OR (95% CI)	*p* Value	β	Wald χ^2^ Value	Adjusted OR (95% CI)	*p* Value
Current-season vaccination behavior	Basic influenza knowledge	0.080	1.468	1.083 (0.953–1.235)	0.226	0.025	0.138	1.026 (0.898–1.172)	0.710	0.043	0.748	1.044 (0.948–1.150)	0.387
	Perceived susceptibility	−0.014	0.135	0.986 (0.917–1.060)	0.713	0.121	8.676	1.128 (1.043–1.225)	0.003	0.064	4.797	1.066 (1.007–1.129)	0.029
	Perceived severity	−0.089	5.442	0.914 (0.847–0.985)	0.020	−0.052	1.827	0.949 (0.880–1.024)	0.177	0.010	0.108	1.010 (0.952–1.072)	0.742
	Perceived vaccination benefit	0.081	2.725	1.084 (0.985–1.194)	0.099	0.094	3.044	1.099 (0.989–1.223)	0.081	−0.003	0.009	0.997 (0.935–1.063)	0.923
	Perceived vaccination barriers	−0.006	0.082	0.994 (0.957–1.034)	0.774	−0.033	1.955	0.968 (0.924–1.013)	0.162	−0.057	10.416	0.944 (0.911–0.977)	0.001
	Action cues	0.195	33.975	1.215 (1.140–1.300)	<0.001	0.192	21.984	1.211 (1.120~1.315)	<0.001	0.133	26.461	1.142 (1.087–1.202)	<0.001
Next-season vaccination intention	Basic influenza knowledge	−0.044	0.306	0.957 (0.818–1.119)	0.580	−0.107	2.885	0.898 (0.793–1.016)	0.089	0.074	3.460	1.077 (0.996–1.165)	0.063
	Perceived susceptibility	0.165	13.278	1.180 (1.081–1.292)	<0.001	0.180	22.936	1.197 (1.113–1.290)	<0.001	0.075	10.123	1.078 (1.029–1.129)	0.001
	Perceived severity	−0.065	2.014	0.937 (0.855–1.024)	0.156	−0.016	0.215	0.984 (0.917–1.054)	0.643	−0.035	2.083	0.966 (0.921–1.012)	0.149
	Perceived vaccination benefit	0.047	0.668	1.048 (0.937–1.174)	0.414	0.105	5.083	1.111 (1.015–1.218)	0.024	0.031	1.511	1.032 (0.982–1.085)	0.219
	Perceived vaccination barriers	−0.020	0.580	0.980 (0.932–1.033)	0.446	−0.024	1.233	0.976 (0.935–1.019)	0.267	−0.042	9.286	0.958 (0.932–0.985)	0.002
	Action cues	0.266	45.504	1.305 (1.212–1.415)	<0.001	0.221	42.155	1.247 (1.169–1.336)	<0.001	0.230	117.971	1.258 (1.208–1.313)	<0.001

## Data Availability

All data used during the study are available from the corresponding author upon request.

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
