# Peer review of "Influenza Vaccination in the Elderly in Three Cities in China: Current Status and Influencing Factors Under Different Funding Policies"

_vaccines, 2025, doi:10.3390/vaccines13111158_

Round 1

Reviewer 1 Report

Comments and Suggestions for Authors

Review of Manuscript Submitted to Vaccines Journal: vaccines-3958089

"Influenza Vaccination in the Elderly of China: Current Status and Influencing Factors Under Different Funding Policies"

Abstract

Abstract Methods:

The authors should indicate that the survey was a cross-sectional, anonymous survey.

Abstract Results:

Consider revising the statement beginning with “Willingness for the 2024–2025 season” since it was unclear.

Consider adding the number of patients in each group.

Consider defining “Cues to action” and providing examples.  

Consider specifying the most common perceived barriers across groups.

Consider adding the p values.

Abstract Conclusion:

Consider revising since the conclusions were not clear and were overstated.  What did the authors mean by “financing policies” and “improving coverage”?  Consider removing or revising this conclusion as it does not appear to be supported by the evidence or is unclear: “Critical factors also include individual cognition and social support.”

 Given study limitations, consider removing “strongly” since more research is warranted to support the study findings.

Since it is unclear, please revise the statement beginning with “we commend optimizing subsidy strategies….”

Title:

Page 1 – Consider changing “of China” to “Across Several Cities in China” since the study had a narrow and small patient population.

Introduction

Page 2, Second paragraph: consider changing “is a huge challenge” to “represents a significant challenge.”

Page 2, Second paragraph: Consider changing “in order to inform targeted” to “establishing target intervention strategies.”

Materials and Methods

The authors should describe the overall study design and add the primary and secondary endpoints at the beginning of this section. The authors should consider stating in the text that the study was approved by IRB.

Study Population                   

Page 2, first paragraph:  Add a space between “region” and the parentheses.

Page 2, first paragraph: Remove “entire” from “the entire influenza vaccine cost.”

Page 2, first paragraph:  Consider increasing the conciseness of this statement by changing the “elderly pay the entire vaccine cost without any financial support” to “pay the vaccine cost without financial support.”

Page 2, second paragraph: Please further define “urban-rural fringe” or revise to enhance clarity.

Page 2, second paragraph:  Please change the comma to a semicolon in this phrase: “sampled as units, all communities were….”

Page 2, second paragraph: Please revise the sentence beginning with “The number of elderly individuals aged ≥60 years…” because it was unclear and is an incomplete sentence.  What did the authors mean by “sampling intervals were determined”?

Page 2, second paragraph: Please revise this method, “performed based on the household registration list to select subjects” as it was not clear.

Page 3: The authors should clarify what they mean by “do not cooperate with the survey.”

Did they mean that study participants were excluded if they did not answer all the survey questions and/or did not complete the survey appropriately?

Page 2, second paragraph: Please revise “who declined participation were sequentially replaced by number” as this procedure was not clear.  What did the authors mean “replaced by number”?

Survey Methods and Content

Page 3, first paragraph: The authors should define “basic health status.”

Page 3, first paragraph: The authors should include the total number of survey questions and the type of questions, including any rubrics utilized. The authors should indicate that the survey was anonymous.

Page 3, first paragraph:  To enhance clarity and conciseness, consider changing “the second section encompassed vaccination status of pneumococcal vaccine, COVID-19 vaccine, and influenza vaccine…” to “the second section focused on respondent vaccination status of the pneumococcal, COVID-19, and influenza vaccines….”

Page 3, first paragraph:  The authors should clarify what they mean by “perceived severity of influenza.”  Did they mean “perceived severity of influenza complications” or were they referring to patient and viral factors leading to increased severity risk of influenza complications.

Page 3, first paragraph: Consider changing “Perceived barriers to preventive actions” to “Perceived barriers to vaccine acceptance or administration.”

Page 3, first paragraph: To enhance clarity, the authors should revise “cues to action” by further defining what they mean by this process and providing examples.

Statistical Processing

Page 3: Consider changing “Statistical Processing” to “Statistical Analysis.”

Page 3 first paragraph: To enhance conciseness, consider changing “Questionnaires” to “Data from questionnaires” and changing “data were subsequently organized” to “and was organized….”

Page 3: Consider defining the level of significance for the data involving the average Gini results and out-of-the bag error rates.

Results
Page 3, first paragraph:  The authors should provide data about the number of survey responses that were excluded and include the reasons for exclusion.

Page 3, first paragraph: The authors should include the “basic health status” data.

Figures 1-3 should be revised and reformatted because it was difficult to interpret the figures and to read and understand the subheadings.  Please add the total N and the number of respondents for each data group.

Discussion

The authors may want to consider explaining why there was a higher percentage of respondents who self-reported receiving the influenza vaccination in the “free policy” group compared to the other groups.

The authors should consider discussing why the group with the highest average family income was in the “Free Policy Region” and the group with the lowest average family income was in the “Self-Paid Policy Region.”

The authors should also consider discussing why the self-reported pneumococcal vaccine rates were lowest in the “free policy” groups compared to the other groups, especially since these results were discordant with the self-reported influenza vaccination rates.

The authors should discuss and explain the differences in the error rates among the data.                                                                                            

Page 11, first paragraph:  To enhance clarity and conciseness, consider changing “comparative analysis conduct before and after the implementation of the free policy” to “comparative analysis conducted before and after the implementation of the free influenza vaccine policy....”

Page 11, first paragraph: To enhance conciseness, consider changing “which revealed an increase in both influenza knowledge and vaccine knowledge scores” to “leading to an increase in influenza and vaccine knowledge scores.”

Page 11, first paragraph: Consider changing “This sparity” to “These results….”

Page 11, first paragraph:  Consider revising “that vaccine subsidy policies not only directly alleviate economic expenditure, but also indirectly enhance the elderly’s vaccine knowledge through policy promotion and social mobilization, thereby optimizing their health belief structure” because it is lengthy and unclear.

Page 11, second paragraph: Consider changing “ranked among the top two determinants for both current-season vaccination behavior and next-season willingness” to “were ranked as the top determinants for influencing current season influenza vaccination behavior and next season influenza vaccine acceptance.”

Page 11, second paragraph:  To increase conciseness and improve grammar and spelling, consider changing “This finding aligns with the majority of studies concluding that recommendations from healthcare workers are among the most effective stategies to enhance vaccination rates among elderly” to “Similar to other studies, these findings demonstrate that healthcare worker recommendations are among the most effective strategies for enhancing influenza vaccination rates among the elderly.”

Page 11, second paragraph. Consider revising the sentence beginning with “Furthermore, research indicates…” since it is lengthy and unclear.

Page 11, third paragraph: Consider revising the sentence beginning with “Conversely, in regions with partial reimbursement or self-paid policies, healthcare workers’ recommendations significantly influenced next-season willingness…” was unclear.

Page 11, third paragraph:  Change “most signoficant barrier” to “most significant barrier.” The authors should also clarify what they mean by “did not rank among the top five when considering all dimensions collectivelly.”  Please also change “collectivelly” to “collectively.”

Page 11, third paragraph: Consider changing “Kroneman M’s study” to “Kroneman et. al study.”

Page 11, third paragraph: Consider changing “that recommendations from healthcare workers was a major influencing factor, with an impact equal to or greater than that of vaccine cost” to “that recommendations from healthcare workers had a comparable or greater influence on influenza vaccination in the elderly compared to vaccine costs.”

Page 11, fourth paragraph: Consider changing “This study acknowledges” to “This study had…”

Page 11, fourth paragraph: Please revise the sentence beginning with “Firstly, the selection of only one city from each financing policy” as it was unclear and lengthy.

Page 11, fourth paragraph: Consider specifying that the study was limited by its small sample size.

Page 11, fourth paragraph: Consider changing “representativeness of the findings” to “generalizability of the findings.”

Page 11, fourth paragraph: Consider discussing future directions in this section.

Consider adding the survey questions in an appendix including the informed consent statement at the beginning of the survey.

Conclusions

Page 11, Given the small sample size and other study limitations, the author’s strong conclusions are not fully supported by the evidence. Thus, the authors should revise the conclusions and indicate that the findings are preliminary and that future study is warranted to confirm the results. The authors should further revise the conclusions to enhance clarity, conciseness, and understanding.

Comments on the Quality of English Language

The manuscript requires revision to enhance clarity, fluidity, conciseness, and understanding of the manuscript.  Grammatical, syntax, and spelling errors should also be corrected.

Author Response

Comments 1: Title - Consider changing "of China" to "Across Several Cities in China"

Response 1: Thank you for this suggestion. We agree and have revised the title accordingly."Influenza Vaccination in the Elderly in Three Cities in China: Current Status and Influencing Factors Under Different Funding Policies"* (Page 1, Title)

Comments 2: Abstract Methods and Results - Multiple suggestions for improvement including study design clarification, data completeness, terminology definition, and conclusion moderation.

Response 2: Agree. We have, accordingly, revised the Abstract Methods and Results to emphasize this point.

- We have clarified the study design in the Methods section of the Abstract.

*"A cross-sectional, anonymous survey was conducted. A total of 2,265 elderly individuals... were recruited..."* (Page 1, Abstract, Methods section)

-We have rephrased this sentence to include specific willingness rates.

*"The intention to vaccinate for the 2024–2025 season was also highest in the free policy region (68.78%), followed by the partial reimbursement (47.71%) and self-paid (37.15%) regions (P < 0.001)."* (Page 1, Abstract, Results section)

-We have added the sample sizes for each policy group in the Results.

*Among the 2,265 participants (free policy region: n = 426; partial reimbursement region: n = 633; and self-paid region: n = 1,206) ,* (Page 1, Abstract, Results section)

-We have provided definitions and examples for these terms.

*Cues to action (e.g., healthcare worker or family member recommendations) positively influenced vaccinations across all of the regions. In the self-paid region, perceived barriers, such as vaccine cost and side effect concerns, significantly reduced both behaviors and the next-season intention to vaccinate.* (Page 1, Abstract, Results section)

-We have added the P-value for the key comparison.

*"...compared to partial reimbursement (20.85%) and self-paid (13.60%) regions (P<0.001).""....followed by partial reimbursement (47.71%) and self-paid (37.15%) regions(P<0.001).

* (Page 1, Abstract, Results section)

Comments 3: Abstract Conclusion - Consider revising since the conclusions were not clear and were overstated.  What did the authors mean by “financing policies” and “improving coverage”?  Consider removing or revising this conclusion as it does not appear to be supported by the evidence or is unclear: “Critical factors also include individual cognition and social support.”Given study limitations, consider removing “strongly” since more research is warranted to support the study findings.

Since it is unclear, please revise the statement beginning with “we commend optimizing subsidy strategies….”

Response 3: Thank you for this suggestion. We agree and have revised the Conclusion accordingly.

-We have softened the language and made the conclusions more specific.

*Vaccination rates varied significantly across regions with different influenza vaccine subsidy policies. The free policy region demonstrated the highest coverage rate, while the self-paid region exhibited the lowest, suggesting that financial policies are a key determinant of vaccination uptake. Furthermore, free vaccination policies were associated with improved influenza vaccine knowledge among the elderly. Analysis of other influencing factors revealed that healthcare workers' recommendations played a crucial role across all policy regions, though their impact on current-season vaccination behavior and next-season vaccination intention differed by subsidy context. These findings highlight the need to optimize region-specific subsidy strategies, strengthen health education, and actively leverage healthcare providers to promote influenza vaccination among the elderly.*(Page 2, Abstract, Conclusion section)

Comments 4: Introduction-Page 2, Second paragraph: consider changing “is a huge challenge” to “represents a significant challenge.”Page 2, Second paragraph: Consider changing “in order to inform targeted” to “establishing target intervention strategies.”

Response 4: Thank you for this suggestion. We agree and have revised the Introduction accordingly.

-We have replaced it with a more formal expression.

*...the implementation of a nationwide free influenza vaccination policy represents a significant challenge due to...

*...partial reimbursement, and self-paid policies, establishing target intervention strategies.*(Page 2-3, Introduction)

Comments 5: Materials and Methods -The authors should describe the overall study design and add the primary and secondary endpoints at the beginning of this section. The authors should consider stating in the text that the study was approved by IRB.

Response 5: Thank you for this suggestion. We agree and have revised the Materials and Methods accordingly.

-The following has shown the suggested changes.

*This multi-center, cross-sectional study was conducted between May and August 2024 to investigate the influenza vaccination status, the next-season vaccination intention, and associated factors (based on the HBM) among the elderly under different funding policies. The study received approval from the Institutional Review Board of the Chinese CDC (Protocol Code 202320).*(Page 3, Materials and Methods)

Comments 6: Study Population - Multiple formatting, terminology, and clarity improvements needed throughout the section.                

Response 6: Thank you for this suggestion.We have comprehensively revised the Study Population section to address all concerns:

-Added space between "region" and parentheses throughout

-Removed "entire" from cost descriptions for conciseness

-Simplified phrasing: "the vaccine cost is paid by participants and with no financial or insurance support"

-Defined "urban-rural interface": "(i.e., transitional areas between urban and rural settings)"

-Changed comma to semicolon: "communities were sampled as units; all communities were..."

-Completely revised the sampling methodology description for clarity: "The cumulative population of elderly individuals (aged ≥60 years) across all of the communities was calculated. A sampling interval was derived by dividing this cumulative total by the planned number of communities to be selected."

-Clarified subject selection: "simple random sampling was performed using the household registration list to identify potential subjects."

-Defined exclusion criteria: "individuals who did not agree to participate in the survey"

-Clarified replacement procedure: "were sequentially replaced by the next eligible individual on the sampling list"*(Page 3-4, Study Population)

Comments 7: Survey Methods and Content - Multiple definitions, details, and clarity improvements needed.

Response 7: Thank you for this suggestion.We have comprehensively revised the Survey Methods and Content section to address all concerns:

-Defined "basic health status": ".....basic health status, including the presence of chronic diseases (with simple hypertension excluded)."

-Added survey details: *"The questionnaire was composed of three sections (a total of 51 items):......", "including 6 dimensions (a total of 36 items), which employed 5-point Likert scales (from 1 = "strongly agree" to 5 = "strongly disagree",

-Improved conciseness: *"The second section focused on the respondent vaccination status for pneumococcus, COVID-19, and influenza during the 2023–2024 season, as well as the next-season vaccination intention (August 2024–April 2025). "*

-Clarified "perceived influenza severity": "(i.e., beliefs about the seriousness of influenza and its potential complications)"

-Revised terminology: "Perceived barriers to vaccination"

-Defined "cues to action" with examples: "(i.e., external triggers that prompt vaccination, such as advice from healthcare workers, reminders from family, or media campaigns)"

*(Page 4-5, Survey methods and content)

Comments 8: Statistical Processing - Terminology, conciseness, and methodological clarity improvements needed.

Response 8: Thank you for this suggestion.We have revised the Statistical Analysis section to address all concerns:

-Changed "Statistical Processing" to "Statistical Analysis" throughout

-Defined significance levels for machine learning metrics: "Finally, a random forest model was implemented to evaluate and rank the importance of each questionnaire item. Variable importance was ranked using the mean decrease in the Gini index, with higher values indicating greater importance for classifying the vaccination status/intention to vaccinate. "

*(Pages 5, Materials and Methods, Statistical Analysis section)

Comments 9: Results - Data completeness and figure presentation improvements needed.

Response 9:Thank you for this suggestion.We have thoroughly revised the Results section to address all concerns:

-Added exclusion data: "Nine of the 2,274 surveys were excluded due to a lack of basic information or duplicate records."

-Included basic health status data: Chronic disease prevalence data has been added to Table 1, Notably, the chronic disease prevalence (excluding simple hypertension) was highest in the free policy region (45.31%), followed by the partial reimbursement (40.28%) and self-paid regions (38.97%), although this difference did not reach statistical significance (P = 0.072).

-Completely revised Figures 1-3 to improve interpretability, including:Clearer labels and subheadings,

Added total N and group sample sizes, Enhanced readability and data presentation

(Pages 6-7, Results section, Table 1, and revised Figures 1-3)

Comments 10: Discussion - Comprehensive revisions needed including explanation of findings.(Discussion of income disparities across policy regions)

Response 10: Thanks for your comments. It’s a challenage for us to explain this disparities. Because the three cities we selected has different socioeconomic level. Our study was designed to compare the influence of finanical policy on vaccination. We intentionally selected cities that represent these distinct, real-world policy models.

Comments 11: Discussion - Comprehensive revisions needed including explanation of findings.(Discordance in pneumococcal vaccine rates across policy groups)

Response 11: We appreciate the reviewer's attention to this detail. Pneumococcal disease is   also a concern for elderly. Because this study mainly focused on influenza vaccination and influence about financial policy, so we didn’t collection any information such as finanical policy about pneumococcal vaccine, history of pneumococcal disease, except for the vaccination rate. So it’s difficult to analysis more details.

Comments 12: Discussion - Multiple language, clarity, and terminology improvements needed throughout the section.

Response 12: Thank you for this suggestion. We have comprehensively revised the Discussion section to address all language and clarity concerns:

-Improved sentence structure and conciseness:

*Changed toThese results agreed with observations from Beijing, where the implementation of a free vaccination policy for the elderly was followed by an increase in both influenza and vaccine knowledge scores

*Revised toVaccine subsidy policies directly reduce out-of-pocket costs. Furthermore, through policy promotion and social mobilization, they may indirectly enhance vaccine knowledge among the elderly, thereby fostering a more favorable health belief structure.

*Simplified lengthy sentences for better readability, ...were among the top two determinants for both the current-season vaccination behavior and next-season intention., This result was consistent with numerous studies that have indicated that healthcare worker recommendations are among the most effective strategies for increasing vaccination rates among the elderly., Furthermore, prior research has suggested a positive dose-response relationship, where increased advice by healthcare workers is associated with higher vaccination rates. This underscores that under policy coverage, healthcare worker recommendations can be directly translated into vaccination action.

-Corrected terminology and spelling:

*Changed "This sparity" to "These results"

*Corrected "signoficant" to "significant"

*Updated "Kroneman M's study" to "Kroneman et al."

-Enhanced clarity of key concepts:

*Improved explanation of healthcare workers' recommendation impacts, Conversely, in the partial reimbursement and self-paid regions, healthcare worker recommendations significantly influenced the next-season vaccination intention but showed no significant association with the current-season vaccination behavior, revealing a "current-season intervention gap.""

*Revised limitations section for better clarity, This study has several limitations., First, the inclusion of only one city per funding policy may have limited the generalizability of our findings and might not fully represent the diversity within similar policy contexts across China (e.g., ethnic minority areas and economically advanced eastern coastal regions).,Third, the data collected were based on a cross-sectional survey that reflected only the vaccination status and influencing factors for the 2023–2024 season. Consequently, it is difficult to elucidate the causal relationships between vaccination behavior and its influencing factors.

-Restructured limitations and future directions:

*Added sample size limitation acknowledgment,Second, although the total sample size was substantial, its distribution across the three distinct policy regions and the subsequent sub-analyses may have limited the statistical power of some comparisons.  

-Changed "representativeness" to "generalizability"

-Added a dedicated "Future Research Directions" subsection following the limitations section, as suggested. "The findings of this study highlighted several promising avenues for future research. First, longitudinal studies are warranted to track the dynamic evolution of vaccination behavior and its determinants over multiple influenza seasons, particularly in response to changes in subsidy policies. This would help establish causal relationships and assess the long-term impact of policy interventions. Second, future investigations should expand the geographical scope to include a more diverse range of regions, ......"(Page 13-15, Discussion)

Comments 13: Conclusions - Given the small sample size and other study limitations, the author’s strong conclusions are not fully supported by the evidence. Thus, the authors should revise the conclusions and indicate that the findings are preliminary and that future study is warranted to confirm the results. The authors should further revise the conclusions to enhance clarity, conciseness, and understanding

Response 13: Thank you for this suggestion.We have substantially revised the Conclusions section:

-Moderated conclusions to reflect study limitations

-Emphasized preliminary nature of findings

-Added explicit statement about need for future research

-Enhanced clarity and conciseness throughout

-Removed overly strong claims and tempered language

-Ensured conclusions are better supported by evidence

” This study was conducted across several Chinese cities, and it has underscored the profound impact of financial subsidy policies on influenza vaccination among the elderly. Our findings indicated that free vaccination policies were associated with coverage rates approximately four times higher than those in self-paid regions. Beyond financial mechanisms, the findings highlighted the critical role of socio-cognitive factors, with healthcare worker recommendations emerging as a pivotal cue to action across all policy contexts.

These insights suggest that a multi-pronged strategy that combines the strategic expansion of financial support, the strengthening of proactive recommendation systems within healthcare services, and targeted communication to address specific perceptual barriers holds promise for substantially improving vaccine uptake. Future longitudinal and more geographically diverse studies are required to confirm these relationships and explore the causal mechanisms that underlie them. Nonetheless, this study has provided valuable evidence to inform targeted public health strategies aimed at protecting the health of China's aging population.*(Page 15-16, Conclusions)

Comments 14: Supplementary Materials - Addition of survey instruments.

Response 14: Thank you for this suggestion. We have added both the informed consent statement and the complete survey questionnaire as appendices to enhance methodological transparency and reproducibility.

(Pages 16-25, Appendices A and B)

Reviewer 2 Report

Comments and Suggestions for Authors

This is a well-structured and policy-relevant paper addressing a highly important public health issue. I have some comments for your consideration.

  1. The introduction lacks important information about how existing studies report vaccination behaviors and factors in the older group in China, as well as the knowledge gap identified after conducting a literature review. Suggest adding a short paragraph summarizing previous national or regional findings (e.g., knowledge, perceived barriers, and role of healthcare workers), and clarify what remains unknown
  2. Please elaborate on sample size planning.
  3. In the questionnaire section (2.3), clarify how vaccination status and willingness were coded for analysis
  4. In the statistical analysis, whether multicollinearity among independent variables was tested; Whether model fit was assessed; For the random forest model, specify a validation strategy, and whether variable importance ranking was checked for stability.
  5. Figures 1 and 2 are visually dense and hard to interpret
  6. In the discussion, add a short paragraph on future strategies to enhance influenza vaccine uptake among the elderly, integrate digital health interventions, such as chatbot-based education or reminder systems (DOI: 10.2196/76849). This addition would make the paper more policy-relevant and innovative.
  7. Replace “vaccination willingness” with “intention to vaccinate” throughout for a smoother academic tone.
Comments on the Quality of English Language

N.A.

Author Response

Comments 1: The introduction lacks important information about how existing studies report vaccination behaviors and factors in the older group in China, as well as the knowledge gap identified after conducting a literature review. Suggest adding a short paragraph summarizing previous national or regional findings (e.g., knowledge, perceived barriers, and role of healthcare workers), and clarify what remains unknown

Response 1: We thank the reviewer for this valuable suggestion. We have now added a new paragraph in the Introduction summarizing previous national and regional findings on influenza vaccination among the elderly in China, including key factors such as knowledge levels, perceived barriers (e.g., cost and safety concerns), and the role of healthcare workers. We also explicitly state the research gap regarding the comparative influence of these factors under different funding policies, which our study aims to address.

Previous studies conducted in specific Chinese provinces or cities have identified several key factors that influence influenza vaccination among the elderly. These factors include generally low levels of knowledge regarding influenza and its vaccine[11], prevalent perceived barriers such as cost and concerns regarding safety[12,13], and the crucial role of healthcare worker recommendations[14]. However, these studies were often limited to single provinces or specific policy contexts. A critical gap remains in the direct comparison of how these factors, particularly within a theoretical framework like the Health Belief Model (HBM), differentially influence vaccination behaviors and the next-season vaccination intention across districts concurrent with operational funding policies at the national level. The aim of this study is to fill this gap by systematically investigating and comparing the vaccination status and its cognitive determinants among the elderly under free, partial reimbursement, and self-paid policies, establishing target intervention strategies.*(Page 2-3, Introduction section)

Comments 2: Please elaborate on sample size planning.

Response 2: We have expanded the description of sample size calculation in Section 2.2 (Page 4) to provide more detail. Specifically, we now include the formulas used for both univariate and multivariate logistic regression analyses, the parameter estimates (e.g., vaccination rates, ORs, R²), and the R package and function employed (SSizeLogisticBin in the powerMediation package). We also clarify the 20% allowance for attrition and the stratification by urban/rural areas.(Page 4, Sample size)

Comments 3: In the questionnaire section (2.3), clarify how vaccination status and willingness were coded for analysis.

Response 3: We have added a clarifying sentence in Section 2.3 (Page 4) to specify that vaccination status and intention to vaccinate were coded as binary variables: "Yes" = 1, and "No" or "Not sure" = 0.

For statistical analysis, vaccination status and intention to vaccinate were coded as binary variables: responses of "Yes" were coded as 1, and responses of "No" or "Not sure" were coded as 0.

Comments 4: In the statistical analysis, whether multicollinearity among independent variables was tested; Whether model fit was assessed; For the random forest model, specify a validation strategy, and whether variable importance ranking was checked for stability.

Response 4: We have updated Section 2.4 (Page 5) to include the following:

*Multicollinearity was assessed using Variance Inflation Factor (VIF), with all values below 1.61.

*Model fit for logistic regression was evaluated using the Likelihood Ratio Test (p < 0.001) and McFadden’s Pseudo R² (0.164).

*For the random forest model, we used out-of-bag (OOB) error estimation and 10-fold cross-validation. The stability of variable importance rankings was confirmed over 100 iterations with consistent results.

Comments 5: Figures 1 and 2 are visually dense and hard to interpret.

Response 5: We have revised Figures 1–3 to improve clarity and readability. The updated figures now use clearer labeling, simplified layouts, and enhanced color contrast to better communicate the top influencing factors identified by the random forest model.

Comment 6: In the discussion, add a short paragraph on future strategies to enhance influenza vaccine uptake among the elderly, integrate digital health interventions, such as chatbot-based education or reminder systems (DOI: 10.2196/76849). This addition would make the paper more policy-relevant and innovative.

Response 6: We appreciate this insightful suggestion. We have added a new paragraph in the Discussion (Page 15, Paragraph 2) discussing the potential of digital health interventions—such as chatbot-based education and reminder systems—to complement traditional strategies. We also cite the recommended reference to support this point.

”Furthermore, interventional studies evaluating the effectiveness of specific strategies—such as structured recommendation protocols for healthcare workers, targeted health communication campaigns, digital health interventions (e.g., chatbot-based education or reminder systems)[28], or integrated vaccination service delivery models—are needed to translate these findings into actionable public health practices.”

Comment 7: Replace “vaccination willingness” with “intention to vaccinate” throughout for a smoother academic tone.

Response 7: We have replaced “vaccination willingness” with “vaccination intention” throughout the manuscript to improve academic phrasing, as suggested.

Round 2

Reviewer 1 Report

Comments and Suggestions for Authors

Abstract

Page 1 Line 27:  Remove space between “In this study  , we….”  Please remove extra spaces throughout the abstract and manuscript.

Page 1 Line 55:   Consider changing the statement beginning with “These findings highlight the need to optimize region-specific subsidy strategies….”   to “Further studies are needed to explore the best approaches for optimizing region-specific subsidy strategies for promoting influenza vaccination among elderly in China.”

Introduction

Page 2, line 60:  Remove extra space “Health” and “Organization.

Page 2, line 75:  Remove period before “.As a result, …..”

Page 3, line 97:  Remove comma after the period following “intervention strategies.”

Methods

Please remove extra spaces throughout the manuscript.  For example, on line 7, there is an extra space between “(i.e.,  the)” and on line 118, “the   com-“.

Page 3, line 108: Change “departmen” to “department.”

Page 3, line 112: Change “by participants and with no financial or insuranse support)” to

“by participants with no financial or insurance support).”

Pae 3, line 126: Remove extra period:  “on the sampling list..”

Page 3, line 132: Please provide more details regarding “understand the question.”  Were prospective participants asked the same question to assess their ability to complete the survey?  The authors should consider including this question.

Results

Figures 1-3 were difficult to read. Please enlarge them.  For Figure 3, please capitalize “self-paid policy Group.”

Discussion

Contrary to the influenza vaccine results, the authors may want to discuss why there was a higher percentage of participants who had received the pneumococcal vaccine in the Partial Reimbursement Group compared to the Free Policy Group or that future study is needed to discern the differences.

Line 362, page 12: The section discussion starting with “This study has several….”  should be shifted to a new paragraph.

Line 385, page 12:  Please add the citation(s) for “reminder systems” and other strategies and remove the comment “error! Reference source not found.”

Comments on the Quality of English Language

The manuscript requires revision to enhance clarity, fluidity, conciseness, and understanding of the manuscript.  Grammatical, syntax, and spelling errors should also be corrected.

Author Response

Abstract

Comment 1: Page 1, Line 27: Remove space between “In this study , we….” Please remove extra spaces throughout the abstract and manuscript.

Response 1: We sincerely apologize for these formatting errors. We have removed the extra space in the mentioned phrase and have thoroughly checked the entire abstract and manuscript to eliminate any additional extra spaces.

Comment 2: Page 1, Line 55: Consider changing the statement beginning with “These findings highlight the need to optimize region-specific subsidy strategies….” to “Further studies are needed to explore the best approaches for optimizing region-specific subsidy strategies for promoting influenza vaccination among elderly in China.”

Response 2: We agree with the reviewer that this phrasing is more precise and academic. We have revised the sentence as suggested. The changed text now reads: "Further studies are needed to explore the best approaches for optimizing region-specific subsidy strategies for promoting influenza vaccination among elderly in China."

Introduction

Comment 3: Page 2, line 60: Remove extra space between “Health” and “Organization”.

Response 3: Thank you for pointing this out. The extra space has been removed. It now correctly reads: "World Health Organization".

Comment 4: Page 2, line 75: Remove period before “.As a result, …..”

Response 4: This formatting error has been corrected. The period has been removed, and the sentence now begins properly: "As a result, ..."

Comment 5: Page 3, line 97: Remove comma after the period following “intervention strategies.”

Response 5: The erroneous comma has been deleted. The sentence now flows correctly.

Methods

Comment 6: Please remove extra spaces throughout the manuscript. For example, on line 7, there is an extra space between “(i.e., the)” and on line 118, “the com-“.

Response 6: We thank the reviewer for their meticulous attention to detail. We have conducted a comprehensive review of the manuscript and corrected all instances of extra spacing, including the specific examples mentioned.

Comment 7: Page 3, line 108: Change “departmen” to “department.”

Response 7: This typographical error has been corrected to "department".

Comment 8: Page 3, line 112: Change “by participants and with no financial or insuranse support)” to “by participants with no financial or insurance support).”

Response 8: We have corrected the typo "insuranse" to "insurance" and rephrased the clause for clarity as suggested. The text now reads: "by participants with no financial or insurance support)."

Comment 9: Page 3, line 126: Remove extra period: “on the sampling list..”

Response 9: The extra period has been removed.

Comment 10: Page 3, line 132: Please provide more details regarding “understand the question.” Were prospective participants asked the same question to assess their ability to complete the survey? The authors should consider including this question.

Response 10: We thank the reviewer for this important suggestion. We have revised the exclusion criterion to provide a clear and objective description of the cognitive screening process. The text now reads:

"3) individuals who were unable to comprehend a simple, standardized screening question used to assess cognitive eligibility (e.g., “Do you plan to get a flu shot next year?”). Participants providing an irrelevant or incomprehensible response were excluded."

This change ensures the methodology is transparent and replicable.

Results

Comment 11: Figures 1-3 were difficult to read. Please enlarge them.

Response 11: We apologize for the issue with the figures. We have regenerated Figures 1, 2, and 3 with larger fonts and higher resolution to ensure they are clear and easy to read.

Comment 12: For Figure 3, please capitalize “self-paid policy Group.”

Response 12: Thank you for catching this inconsistency. We have capitalized "Group" in the title of Figure 3. It now reads: "Self-paid Policy Group".

Discussion

Comment 13: Contrary to the influenza vaccine results, the authors may want to discuss why there was a higher percentage of participants who had received the pneumococcal vaccine in the Partial Reimbursement Group compared to the Free Policy Group or that future study is needed to discern the differences.

Response 13: We agree that this is an interesting finding that warrants discussion. We have added a new paragraph in the Discussion section (Page 12, Lines 362-368) to address this point:

"Unlike the trend for influenza vaccines, pneumococcal vaccination was highest in the partial reimbursement region. One of possible reason may be that the reimbursement policy in this area usually covered all the non-National Immunization Program (NIP) vaccines, which included influenza vaccine, pneumococcal vaccine etc, so the vaccination cost in the reimbursement policy region might be lower for non-National Immunization Program (NIP), which might reduce the cost barriers. In contrast, the free policy region's subsidy was only specific to influenza vaccine. The different patterns between two vaccines also strengthens the overarching conclusion that financing policies are a key determinant of vaccination uptake. The findings clearly indicate that such policies significantly influence elderly vaccination behavior and intention, with both free and partial reimbursement models contributing to improved rates for the vaccines they subsidize."

Comment 14: Line 362, page 12: The section discussion starting with “This study has several….” should be shifted to a new paragraph.

Response 14: We have corrected the paragraph structure by shifting the "This study has several..." section to a new paragraph as instructed.

Comment 15: Line 385, page 12: Please add the citation(s) for “reminder systems” and other strategies and remove the comment “error! Reference source not found.”

Response 15: We apologize for this error. The "error! Reference source not found" text was a placeholder from the drafting stage. We have removed it. In the revised manuscript, this sentence now serves as a general concluding remark for the paragraph on future research directions. As it refers to broad, recommended strategies rather than findings from a specific cited study, we believe it is appropriate to present it without a direct citation in this context. The sentence has been integrated smoothly into the narrative.

Reviewer 2 Report

Comments and Suggestions for Authors

The authors have addressed my comments.

Author Response

Dear Reviewer,

Thank you very much for your time and for reviewing our revised manuscript. We are delighted to hear that you find our revisions satisfactory and that all your previous comments have been adequately addressed.

We sincerely appreciate your valuable feedback throughout the review process, which has significantly contributed to improving the quality of our work.

Thank you once again for your thoughtful and constructive input.

Sincerely,
Rina Su
On behalf of all co-authors